# Unveiling the Catalytic Roles of *DsBBS1* and *DsBBS2* in the Bibenzyl Biosynthesis of *Dendrobium sinense*

**DOI:** 10.3390/molecules29153682

**Published:** 2024-08-03

**Authors:** Liyan Liu, Huiyan You, Lixuan Ye, Qiongjian Ou, Ying Zhao, Jia Wang, Jun Niu

**Affiliations:** Key Laboratory of Genetics and Germplasm Innovation of Tropical Special Forest Trees and Ornamental Plants—Ministry of Education, School of Tropical Agriculture and Forestry, Hainan University, Haikou 570228, China; lly13101@163.com (L.L.); wybplyx666666@163.com (H.Y.); xiaoyetongxue6@outlook.com (L.Y.); oqj1219@163.com (Q.O.); zhaoying3732@163.com (Y.Z.)

**Keywords:** *Dendrobium sinense*, bibenzyl compounds, bibenzyl synthase, enzyme activity

## Abstract

*Dendrobium sinense*, an endemic medicinal herb in Hainan Island, is rich in bibenzyl compounds. However, few studies have explored the molecular mechanisms of bibenzyl biosynthesis. This study presents a comprehensive analysis of DsBBS1 and DsBBS2 function in *D. sinense*. A molecular docking simulation revealed high-resolution three-dimensional structural models with minor domain orientation differences. Expression analyses of *DsBBS1* and *DsBBS2* across various tissues indicated a consistent pattern, with the highest expression being found in the roots, implying that they play a pivotal role in bibenzyl biosynthesis. Protein expression studies identified optimal conditions for DsBBS2-HisTag expression and purification, resulting in a soluble protein with a molecular weight of approximately 45 kDa. Enzyme activity assays confirmed DsBBS2’s capacity to synthesize resveratrol, exhibiting higher V_max_ and lower K_m_ values than DsBBS1. Functional analyses in transgenic *Arabidopsis* demonstrated that both DsBBS1 and DsBBS2 could complement the *Atchs* mutant phenotype. The total flavonoid content in the *DsBBS1* and *DsBBS2* transgenic lines was restored to wild-type levels, while the total bibenzyl content increased. DsBBS1 and DsBBS2 are capable of catalyzing both bibenzyl and flavonoid biosynthesis in *Arabidopsis*. This study provides valuable insights into the molecular mechanisms underlying the biosynthesis of bibenzyl compounds in *D. sinense*.

## 1. Introduction

The *Dendrobium* plants, the second largest within the Orchidaceae family, thrive in warm, humid, and semi-shaded conditions [1]. This genus is widely distributed across the tropical and subtropical regions of Asia–Pacific, extending to Australia and the Americas [1,2]. In China, *Dendrobium* boasts a rich germplasm resource with approximately 110 species, which are primarily distributed in southwestern regions including Guangdong, Guangxi, Hainan, and Taiwan [3]. The *Dendrobium* genus possesses significant ecological and cultural importance as well as considerable economic value [4]. Historically, the *Dendrobium* genus has been highly valued for its diverse pharmacological properties in China and Southeast Asia. *Dendrobium* plants are abundant in bioactive compounds, such as phenols, polysaccharides, alkaloids, bibenzyls, and flavonoids [2]. Pharmacological studies revealed their anti-aging, anti-microbial, antioxidant, and anti-tumor effects [5].

*Dendrobium sinense* is indigenous to Hainan Province and is primarily found in the tropical mountain rainforest of the central and western regions of Hainan Island [6]. This plant occupies a significant role in traditional Chinese medicine and has attracted increasing attention because of its diverse chemical constituents and potential pharmacological effects [7]. A previous study identified bibenzyl as a key secondary metabolite in *D. sinense*, demonstrating substantial inhibitory effects on various human cancer cell lines [8]. Although our prior research identified a key bibenzyl synthase (BBS) enzyme in *D. sinense* [9], its underlying biosynthetic mechanisms are not yet fully understood.

Bibenzyl compounds, a class of polyketide molecules, have been extensively extracted and identified from the *Dendrobium* genus, with approximately 89 derivatives identified from 46 species [10]. These small molecules, including erianin, moscatilin, and gigantol, have the capacity to inhibit cell proliferation, migration, and invasion and induce apoptosis, thereby contributing to tumor progression [11,12,13]. Bibenzyl compounds have demonstrated efficacy in alleviating symptoms of diabetes [14]. Additionally, these compounds exhibit neuroprotective properties [15]. Furthermore, bibenzyl compounds have been shown to inhibit the production of pro-inflammatory cytokines and reduce inflammation in cellular models [16]. The antioxidant properties of bibenzyl compounds are also noteworthy, with several derivatives exhibiting superior activity compared to antioxidants such as vitamin C [17]. Bibenzyl compounds from the *Dendrobium* species represent a rich source of bioactive natural products, yet the biosynthesis of bibenzyl compounds is a complex process. Polyketide synthases (PKSs) catalyze the formation of the polyketide backbone, which is a rate-limiting step in the biosynthesis of diverse polyketide compounds [18].

Type III PKSs are a supergene family of enzymes that play a pivotal role in the biosynthesis of diverse polyketide structures, including but not limited to chalcones, pyrones, acridones, phloroglucinols, stilbenes, and resorcinolic lipids [19]. Currently, chalcone synthase (CHS) is well studied for its role in flavonoid biosynthesis, and it contributes to the production of over 6000 naturally occurring flavonoids [20]. In addition to CHS, plants have also revealed an increasing number of functionally diverse CHS-like type III PKSs, including benzalacetone synthase (BAS), styrylpyrone synthase (SPS), *p*-coumaroyl triacetic acid synthase (CTAS), and stilbene synthase (STS) [18,19]. Type III PKSs have been engineered to synthesize novel polyketide molecules through precursor-directed and structure-based mutagenesis approaches [21]. Notably, BBS, a member of the type III PKS family, is the rate-limiting enzyme in the accumulation of bibenzyl compounds, but there are currently limited studies on this enzyme [9,22].

The high-throughput sequencing of *D. sinense* identified a total of ten type III *PKS*s, including two *DsBBS* genes [20]. To evaluate the biological activity of DsBBS proteins, a molecular docking analysis was performed to simulate the docking of substrates with the proteins. Subsequently, an RT-qPCR analysis was conducted on *D. sinensis* roots, pseudobulbs, and leaves to compare the expression patterns of the *DsBBS1* and *DsBBS2* genes. The enzyme activity of DsBBS1 related to bibenzyl biosynthesis has been previously reported [9], and therefore, this study focused on determining the in vitro activity of DsBBS2. To compare their in vivo activity, *Arabidopsis* mutants (AT5G13930, SALK_076535C) were selected for a transgenic analysis. A comparative analysis of DsBBS1 and DsBBS2 enzyme activities, both in vitro and in vivo, contribute to clarifying their respective roles and distinguishing their differences. These findings provide a basis for the further elucidation of the molecular mechanisms underlying bibenzyl biosynthesis.

## 2. Results

### 2.1. Molecular Docking Simulation of DsBBS1 and DsBBS2

The three-dimensional structural models of DsBBS1 and DsBBS2 were modeled using Phyre2 (Figure 1). A quality evaluation indicated that 99.4% of the amino acids from both the DsBBS1 and DsBBS2 models fall within the allowed regions, implying that they are high-quality prediction models (Appendix A). A comparison of the three-dimensional structures of DsBBS1 and DBBS2 showed a small RMSD value of 0.14. The two structures overlap almost completely except for subtle differences in the domain orientation of one *α*-helix, two *β*-sheets, and some random curls (Figure 1).

To mimic the natural interaction of a ligand with the protein, the AutoDock Tools (ADT) program was adopted for molecular docking. As for the *p*-coumaryl-CoA ligand, although the number of hydrogen bonds in the DsBBS2 protein was higher than that in the DsBBS1 protein, they both contained aspartate at position 136 (ASP-136) (Figure 1). Similarly, the number of hydrogen bonds between the DsBBS2 protein and malonyl-CoA was higher than that in the DsBBS1 protein, but they shared the ASP-136 and leucine at position 137 (LEU-137) (Figure 1). Interestingly, ASP-136 was found in all docking models.

### 2.2. Expression Analysis of DsBBS1 and DsBBS2 in Different Tissues

To better understand the function of the *DsBBS1* and *DsBBS2* genes, a comparative analysis of the gene expression levels was conducted in *D. sinense* roots, pseudobulbs, and leaves. The *DsBBS1* and *DsBBS2* genes showed a similar expression profile in different tissues, exhibiting the highest expressions in the roots, followed by the pseudobulbs and leaves (Figure 2). This concordance in expression patterns suggested a possible shared or complementary function between the two genes and a potentially high demand for bibenzyl biosynthesis in *D. sinense* roots.

### 2.3. Protein Expression of DsBBS2

The activity of the DsBBS1 protein was confirmed in our previous research [9]. Consequently, this study aimed to evaluate the activity of DsBBS2. To optimize the conditions for high-quality DsBBS2 protein expression, a screen of expression and solubility conditions was required. The recombinant DsBBS2-HisTag was induced by 0.1, 0.3, 0.5, 0.8, and 1.0 mmol of isopropyl-β-D-thiogalactopyranoside (IPTG). It was evident that the DsBBS2-HisTag protein was poorly expressed without induction (0 mmol IPTG), yet high yields were achieved upon induction in all tested solubility conditions (Figure 3a). Notably, the highest protein content was achieved with 1.0 mmol/L IPTG induction. Moreover, the expression conditions of the DsBBS2-HisTag protein were optimized at varying temperatures and induction conditions. The DsBBS2-HisTag protein predominantly formed a precipitate at 37 °C, while it was predominantly found in the supernatant at 15 °C (Figure 3b). Therefore, the conditions of 1.0 mmol of IPTG, 15 °C, and 24 h are optimal for the purification of soluble DsBBS2-HisTag protein.

The DsBBS2-HisTag protein was successfully purified using HisTag affinity chromatography. The SDS-PAGE analysis revealed that the DsBBS2-HisTag protein had a molecular weight of approximately 45 kDa (Figure 3c), which aligns with the theoretical value of 42.79 kDa [20]. Protein standards were used with the bicinchoninic acid (BCA) reagent to generate a standard curve. Utilizing the standard curve, the concentrations of the purified DsBBS2-HisTag proteins were determined. The highest concentration achieved for the recombinant DsBBS2 protein was 0.8 μg/μL. These samples are suitable for a subsequent analysis of in vitro enzyme activity.

### 2.4. Enzyme Activity Analysis of DsBBS2

To clearly understand the nature of DsBBS2 and its activity in vitro, the purified recombinant protein was incubated with the substrates. The HPLC results show that one product shared the same retention time with resveratrol (Figure 4a), indicating that DsBBS2 could use the substrate of *p*-coumaroyl-CoA to produce resveratrol. The kinetic parameters of the purified DsBBS2 were determined by measuring the enzyme activity with *p*-coumaroyl-CoA at different concentrations. The V_max_ of the recombinant DsBBS2-HisTag protein for resveratrol production was 1.62 ± 0.10 pmol/s·mg, and the K_m_ value was 2.05 ± 0.31 mmol/L (Figure 4b). Under the same experimental condition with DsBBS1 [9], the V_max_ and K_m_ values of DsBBS2 were significantly higher and lower, respectively, compared with the kinetic parameters for DsBBS1 (Figure 4c).

### 2.5. Functional Analysis of DsBBS1 and DsBBS2 in Transgenic Arabidopsis

To further verify whether there is a difference in the activity between DsBBS1 and DsBBS2 in vivo, *Arabidopsis* (Columbia ecotype) was chosen for a transgenic analysis. A homology comparison with Araport11 protein sequences indicated that AtCHS (AT5G13930.1) protein was the one with the highest homology (*E*-value = 0) to the DsBBS1 and DsBBS2 proteins. Thus, the homozygous *Atchs* mutant (SALK_076535C) that was identified by the three-primer method was used for further studies. Through resistance screening and PCR validation, the homozygous transgenic lines of *DsBBS1* and *DsBBS2* were used for functional analysis (Appendix A).

To understand the possible effects of the *DsBBS* genes on *Arabidopsis* growth and development, wild-type and *Atchs* mutant plants were used as controls to observe the differences in developmental phenotypes. During the vegetative growth stage, the growth of *Atchs* mutants was significantly weaker than that of the wild type (Figure 5a). *DsBBS1* and *DsBBS2* complementation rescued the *Atchs* mutant phenotype (Figure 5a). Interestingly, it was observed that the bolting time of the wild type was the shortest, followed by *DsBBS1* and *DsBBS2* mutant complementation, and the mutant type (Figure 5b). Moreover, the mutant pods were smaller than the wild-type pods and *DsBBS1* transgenic lines, whereas the *DsBBS2* transgenic lines had the longest pod length (Figure 5c). There was no significant difference in the seed size among the different plants (Figure 5d).

To explore the effects of *DsBBS1* and *DsBBS2* expression on polyketide biosynthesis, the total bibenzyl and flavonoid contents underwent a comparative analysis. The results show that the total bibenzyl content was 0.76% and 0.72% in the wild type and mutants, repetitively (Figure 6). However, the total flavonoid content was 1.76% in the wild type, which was significantly higher than the 0.27% value found in the *Atchs* mutants. This finding clearly shows that *AtCHS* is only involved in the accumulation of flavonoids. Compared with mutant plants, the heterologous expressions of the *DsBBS1* and *DsBBS2* genes only led to increases in the total bibenzyl content by 66.02% and 32.53%, respectively (Figure 6). Unexpectedly, *DsBBS1* and *DsBBS2* expressions both led to the recovery of the flavonoid content in the *Atchs* mutant, exhibiting nearly the same activity (Figure 6). These results strongly suggest that the heterologous expressions of *DsBBS1* and *DsBBS2* in *Arabidopsis* successfully compensated for the decrease in the total flavonoid content caused by the loss of *Atchs. DsBBS1 and DsBBS2* could catalyze bibenzyl and flavonoid biosynthesis in *Arabidopsis.*

## 3. Discussion

*Dendrobium* species are a significant source of bibenzyls. To date, 89 bibenzyl derivatives have been extracted and identified from 46 *Dendrobium* species [10]. Bibenzyl compounds have a variety of pharmacological activities, such as anti-tumor activity [23], neuroprotective effects [24], antioxidant effects [25], anti-inflammatory activity [16], and antibacterial and antiviral effects [26]. Considering their multifaceted pharmacological profiles, bibenzyls hold promise for the development of new drugs to treat a variety of conditions. The elucidation of key enzymes involved in bibenzyl biosynthesis are paramount for the enhancement of their biosynthetic efficiency, potentially through metabolic engineering strategies. However, there is little research on the key enzymes involved in the biosynthesis of bibenzyl compounds in *Dendrobium* species, especially in *D. sinense*.

Type III PKSs specialize in synthesizing specific types of polyketides, such as stilbenes, flavonoids, and bibenzyls, which possess a wide range of biological activities and are predominantly found in plants [18,27]. Type III PKSs are the first committed enzyme in the biosynthesis of all polyketides. The BBS protein is the rate-limiting enzyme for bibenzyl biosynthesis [22]. Due to the lack of genomic information for *D. sinense*, the mining of type III PKS genes was performed through transcriptome sequencing. When we performed transcriptome sequencing for the first time using the Illumina platform, only one *DsBBS* gene was identified, namely DsBBS1 [9]. Subsequently, we combined PacBio third-generation sequencing technology with Illumina sequencing data to reanalyze the type III PKS genes in *D. sinense*, resulting in the discovery of 10 type III *PKS* genes, including 7 *DsCHS* genes, 2 *DsBBS* genes, and 1 *DsPKS* gene [20]. This is the most comprehensive analysis of type III PKSs in *D. sinense* reported so far.

A multiple alignment analysis indicated a 95.10% similarity between the DsBBS1 and DsBBS2 protein sequences [20]. Indeed, the three-dimensional structural models of DsBBS1 and DsBBS2 were almost identical (Figure 1). The similarity in protein structure between DsBBS1 and DsBBS2 implies that they likely have similar functions, catalyzing the formation of bibenzyl compounds [19,28]. Nevertheless, no matter which substrate (malonyl-CoA or *p*-coumaryl-CoA) is applied, the number of hydrogen bonds in the molecular docking results for the DsBBS2 protein was higher than that for the DsBBS1 protein. This is mainly due to differences in the non-conserved amino acid residues near the catalytic center. There is a growing body of evidence that optimal hydrogen bonding in enzyme active sites may offer a significant contribution to catalytic accelerations [29]. Thus, DsBBS1 and DsBBS2 have the same catalytic activity, but their catalytic efficiency may vary.

To further explore the functional differences between *DsBBS1* and *DsBBS2*, the expression levels of *DsBBS1* and *DsBBS2* were detected in different tissues. The resulting expression profiles of the *DsBBS1* and *DsBBS2* genes were same, with a significantly higher expression in the roots (Figure 2). A previous study showed that *D. sinense* roots had a higher bibenzyl content than the pseudobulbs and leaves [9]. The positive relationship between the gene expression levels and bibenzyl content suggests that *DsBBS1* and *DsBBS2* are key genes for bibenzyl biosynthesis and accumulation in roots. It is difficult to tell from these expression results whether there are functional differences between the two *DsBBS* genes.

Therefore, to further explore the differences between their enzyme activities, the recombinant protein was obtained through prokaryotic protein expression and chromatography purification. The kinetics of in vitro DsBBS1 enzyme activity has been carried out [9], and thus, we carried out the present study on the DsBBS2 enzyme activity. Interestingly, the V_max_ and K_m_ values of the recombinant DsBBS2 protein were higher and lower than those of DsBBS1 protein, respectively [9]. These findings imply that when using malonyl-CoA and *p*-coumaryl-CoA as substrates, the recombinant DsBBS2 protein showed higher catalytic efficiency than DsBBS1 [30]. It was reported that enzyme–substrate interactions were driven by the stereospecific binding of the substrate to the active site of the enzyme [31]. One limitation of this in vitro enzyme activity measurement is that only one common substrate (*p*-coumaryl-CoA) of type III PKSs was studied. The preferred physiological substrate for DsBBS1 and DsBBS2 needs further verification.

To confirm the *DsBBS1* and *DsBBS2* functions in vivo, we generated their transgenic plants for functional studies. A total of four type III *PKS* genes have been identified in *Arabidopsis* [32], but no *BBS* gene was found. A homology comparison showed that the AtCHS (AT5G13930.1) protein was the one with the highest homology both to DsBBS1 and DsBBS2. Thus, the homozygous *Atchs* mutant was selected for transgenic studies. The *AtCHS* has been identified as the only gene participating in flavonoid biosynthesis [33]. Indeed, the total flavonoid content clearly dropped in the *Atchs* mutant, but there was no difference in the bibenzyl content between the mutant-type and wild-type plants (Figure 6). Despite AtCHS having a high homology with DsBBS1 and DsBBS2, the loss of *AtCHS* function only affects flavonoid accumulation.

It is apparent that a significant increase in the total bibenzyl content was observed both in the *DsBBS1* and *DsBBS2* transgenic lines compared with the wild type (Figure 6). The findings indicate that heterologous expressions of *DsBBS1* and *DsBBS2* can increase the bibenzyl content, which is consistent with the findings of a previous report [34]. Interestingly, the expressions of heterologous *DsBBS1* and *DsBBS2* genes both can compensate for the loss of *Atchs* and restore the total flavonoid content (Figure 6). This result may be explained by the fact that the biosynthesis of bibenzyls and flavonoids uses the same substrates, and all type III PKSs share a common three-dimensional overall fold [28]. These characteristics can lead to complementary functions between *AtCHS* and *DsBBS.* It is also worth noting that there was no significant difference in the total flavonoid content or bibenzyl content between the *DsBBS1* and *DsBBS2* transgenic lines (Figure 6). In contradiction to the results of the in vitro enzyme activity assay, there was no significant difference in the enzyme activity between the heterologous expressions of *DsBBS1* and *DsBBS2* in *Arabidopsis*. The inconsistency may stem from differences between in vitro and in vivo conditions; for instance, plant cells might offer an optimal physiological environment and substrate for the activity of DsBBS1 and DsBBS2. Additionally, variations in protein processing, modification, and folding across distinct expression systems can affect enzyme activity [35].

## 4. Materials and Methods

### 4.1. DsBBS Homology Modeling and Molecular Docking

The protein sequences of DsBBS1 (WHE45950.1) and DsBBS2 (WHE45951.1) were downloaded from the NCIB database [20]. The three-dimensional structures of the proteins were predicted using the Phyre2 server (default parameters). Visualization of the three-dimensional structures was performed using PyMOL 2.52.4, with the alignment module being employed to compare structural differences between the two proteins. The SAVES online server (https://saves.mbi.ucla.edu/, accessed on 10 December 2023) was utilized to assess the reliability of the predicted model. The three-dimensional structures of *p*-coumaroyl-CoA and malonyl-CoA were obtained from the Pubchem database (pubchem.ncbi.nlm.nih.gov/, accessed on 10 December 2023). Using ADT 4.2.6, semi-flexible docking of the two DsBBS proteins with *p*-coumaroyl-CoA and malonyl-CoA was conducted. The ligand was set to 17 twistable keys, and the binding pocket size parameters were set to X:126, Y:126, and Z:106. The coordinates of the search space center were X center: 26.535, Y center: 54.225, and Z center: 54.353, and simulation docking was repeated 50 times using a genetic algorithm. The docking results were visualized in PyMOL, and hydrogen bonds were plotted in residue mode.

### 4.2. RT-qPCR Analysis

Total RNA was isolated from the roots, pseudobulbs, and leaves of *D. sinense* using the RNA Easy Plant Tissue Kit (Taingen, Beijing, China). Subsequently, cDNA was synthesized from the total RNA using the High-Capacity cDNA Archive Kit (Thermo Fisher Scientific, Shanghai, China). RT-qPCR was conducted using the MonAmpTM ChemoHS qPCR Mix Kit (Monad, Guangzhou, China) on a LightCycler 96 System (Roche). Based on our previous report, *ADF11* and *ACBP2* were used as the reference genes across various tissues of *D. sinense* [36]. Details of the RT-qPCR primers are provided in Appendix A. The RT-qPCR primers are shown in Appendix A. Expression levels were calculated using the 2^−ΔΔCt^ method. Reactions were performed in triplicate.

### 4.3. DsBBS2 Cloning and Protein Purification

The cloning of *DsBBS2* genes was achieved using the primers DsBBS2-F and DsBBS2-R (Appendix A). The resulting PCR products were then cloned into PCloneEZ-TOPO (Solarbio, Beijing, China), and sequencing was carried out by Sangon, Guangzhou, China.

The pET28a vector underwent digestion with *Eco*RI and *Sac*I (NEB, Beijing, China). Specially designed primers of DsBBS2-HisTag-F and DsBBS2-HisTag-R included the terminal homologous sequence of the pET28a vector (15 bp), restriction enzyme cutting sequence, and the gene amplification primer sequence (Appendix A). Utilizing the ClonExpress^®^ II One Step Cloning Kit (Vazyme, Nanjing, China), the *DsBBS2* gene was seamlessly integrated into the linearized pET28a vector to produce pET28a-*DsBBS2*. Subsequently, the pET28a-*DsBBS2* vector was introduced into the *Escherichia coli* strain BL21(DE3) for the purpose of protein expression. Protein expression and purification were carried out according to previous methods [9,20]. The purity of the DsBBS2-HisTag protein was assessed through SDS-PAGE, and the protein bands were visualized after staining with Coomassie brilliant blue.

### 4.4. Enzymatic Assay

To assess the enzymatic activity of the DsBBS2 protein, an in vitro enzyme activity assay was carried out. The enzymatic reaction mixture, including 10 μg DsBBS-HisTag protein, 50 mM Hepes buffer (Bioshorp, Beijing, China), and 0.5 mM Malonyl-CoA lithium salt (Yuanye, Shanghai, China), was prepared. During kinetic studies, the enzymatic reaction rate of DsBBS2 was measured for different concentrations of *p*-coumaroyl CoA (Yuanye, Shanghai, China), including 0.5, 1.0, 3.0, and 5.0 mM. This mixture was incubated at 37 °C for 30 min to allow for enzymatic reaction. Subsequently, 50 μL of methanol was added to halt the reaction. After centrifugation at 12,000 rpm for 10 min, a 20 μL aliquot of the supernatant was analyzed using high-performance liquid chromatography (HPLC) to determine the production content.

### 4.5. HPLC Analysis

The samples were analyzed using an LC-100 PUMP system (Wufeng, Shanghai, China) paired with an HC-C18 column (18 μm, 4.6 × 250 mm, supplied by Agilent, Santa Clara, CA, USA). The chromatographic conditions were set as follows: a column temperature of 30 °C, an injection volume of 20 μL, solution A consisting of 0.1% phosphate water, and solvent B being 100% acetonitrile. The flow rate was maintained at 1 mL/min. The gradient elution program for the solvents was as follows: from 0 to 6 min, 70% solution A and 30% solvent B were used; from 6 to 40 min, there was a gradual shift to 55% solution A and 45% solvent B. The HPLC data were estimated based on the calibration curve established with resveratrol standard (Yuanye, Shanghai, China).

### 4.6. Genetic Transformation in Arabidopsis

The T-DNA insertion mutants of *Atchs* (AT5G13930 and SALK_076535C) were purchased from AraShare (www.arashare.cn/index/, accessed on 13 September 2021). Three-primer method was used to screen for homozygous mutations (Appendix A), which were used for genetic transformation. The linearization of the pBI121 vector was conducted by *Bam*HI and *Sma*I (Tolobio, Beijing, China). Using specially designed primers (Appendix A), the *DsBBS1* and *DsBBS2* genes were seamlessly integrated into pBI121 to produce pBI121-*DsBBS1* and pBI121-*DsBBS2* using the ClonExpress^®^ II One Step Cloning Kit (Vazyme, Nanjing, China), respectively. The pBI121-*DsBBS1* and pBI121-*DsBBS2* vectors were transformed into *Agrobacterium tumefaciens* GV3101. The floral dip method was used for genetic transformation in *Arabidopsis*. The T3 homozygous transgenic lines were screened as previously described [20].

### 4.7. Determination of Total Bibenzyl and Flavonoid Contents

Three-month-old *Arabidopsis* seedlings were used as the material to collect leaves for the analysis of the total bibenzyl and flavonoid contents according to a previous method [20,22]. Briefly, the freeze-dried sample was refluxed with anhydrous ethanol at 90 °C for 2 h, followed by centrifugation at 12,000 rpm for 10 min. The supernatant was then concentrated and mixed with 10 mL methanol for the analysis of the total bibenzyl content. As for the determination of total flavonoid content, the concentrated supernatant was transferred into a volumetric flask (25 mL), and anhydrous ethanol was added to obtain a final volume of 25 mL. The assay involved the addition of 30% ethanol and reagents such as 5% sodium nitrite and 10% aluminum nitrate, with subsequent reactions induced by 1 mol/L NaOH. Gigantol (Yuanye, Shanghai, China) was used for generating standard curves to analyze the total bibenzyl content, while rutin (Yuanye, Shanghai, China) was used for the total flavonoid content. Absorbance of the bibenzyl and flavonoid compounds was measured at 280 and 510 nm, respectively. The assay was performed in biological triplicate with three technical replicates each time. The ANOVA analysis was conducted using GraphPad Prism 8.0.2 (*p* < 0.05).

## 5. Conclusions

This study has elucidated the functional roles of the *DsBBS1* and *DsBBS2* genes in the biosynthesis of bibenzyl compounds in *D. sinense*. The molecular docking analysis and gene expression profiles indicated that both genes play a significant role in the production of bibenzyls. The optimization of DsBBS2-HisTag expression conditions and the subsequent enzyme activity analysis revealed that DsBBS2 was able to catalyze the formation of bibenzyl compounds. A functional analysis in *Arabidopsis* demonstrated that both *DsBBS1* and *DsBBS2* can compensate for the loss of *Atchs*, restoring the total flavonoid content to wild-type levels. Moreover, the heterologous expressions of the *DsBBS1* and *DsBBS2* genes led to an increase in the total bibenzyl content. These findings enhance our understanding of the molecular mechanisms underlying bibenzyl biosynthesis in *D. sinense*.

## Figures and Tables

**Figure 1 molecules-29-03682-f001:**
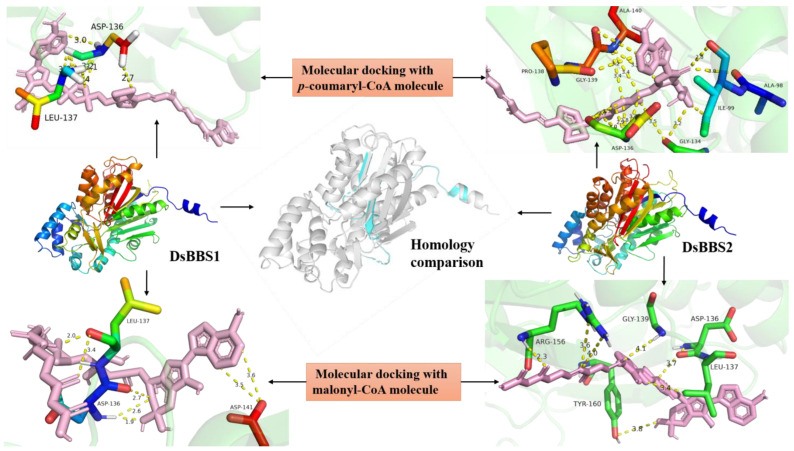
Mimetic molecular docking of DsBBS proteins. Three-dimensional structural models of the DsBBS1 and DsBBS2 proteins were created using the Phyre2 server. The three-dimensional structures of DsBBS1 and DBBS2 were compared using PyMOL. The overlapping parts of the two three-dimensional structures are marked in gray, and the differences are marked in blue. The RMSD between the two structures is 0.14. The molecular docking of DsBBS1 and DBBS2 with *p*-coumaryl-CoA and malonyl-CoA was performed using ADT 4.2.6. Dotted yellow lines represent hydrogen bonds. The specific number on the bond indicates bond length, and the unit is Å.

**Figure 2 molecules-29-03682-f002:**
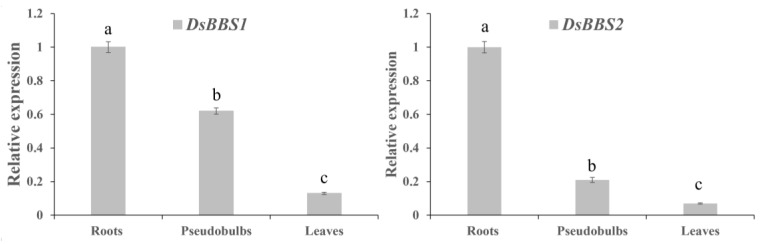
The relative expression levels of *DsBBS1* and *DsBBS2* in different *D. sinense* tissues. *ADF11* and *ACBP2* were used as the reference genes. The expression levels were calculated using the 2^−ΔΔCt^ method. Different letters indicate significant difference (*p* < 0.05; ANOVA).

**Figure 3 molecules-29-03682-f003:**
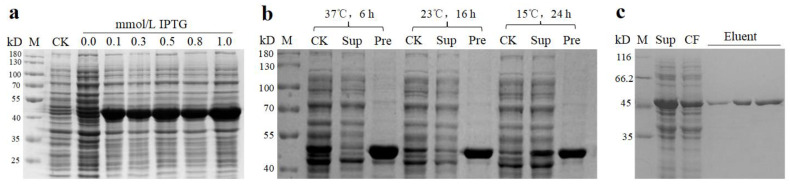
The protein purification of DsBBS2-HisTag protein. (**a**) The optimum inducement concentration of IPTG. (**b**) The optimum inducement time and temperature. (**c**) An electrophoretic diagram of purified DsBBS protein. Abbreviations: CF, cleaning fluid; CK, control check; IPTG, isopropyl-β-D-thiogalactopyranoside; M, marker; Pre, precipitate; Sup, supernatant.

**Figure 4 molecules-29-03682-f004:**
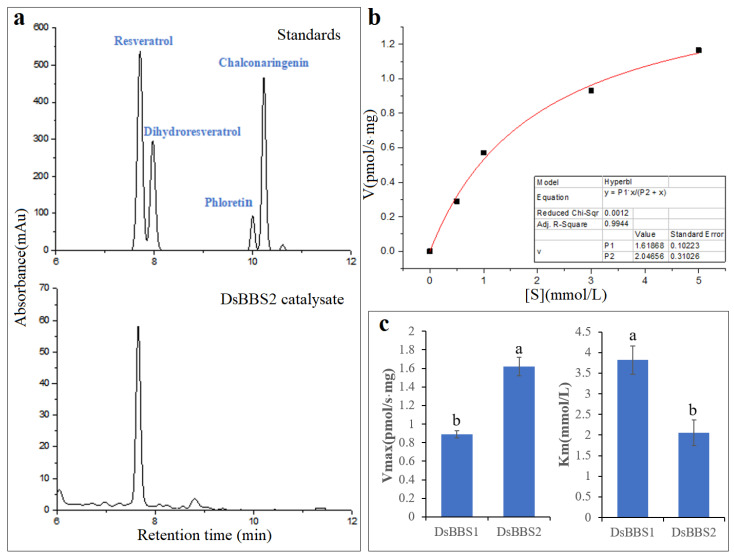
The in vitro enzyme assays of DsBBS2. (**a**) HPLC chromatograms of the reaction productions. The above picture shows the standards, and the below shows the productions by DsBBS2. (**b**) The nonlinear curve fitting of DsBBS2 activity. (**c**) A comparative analysis of the V_max_ and K_m_ values between DsBBS1 and DsBBS2. Different letters indicate significant difference (*p* < 0.05; *t*-test).

**Figure 5 molecules-29-03682-f005:**
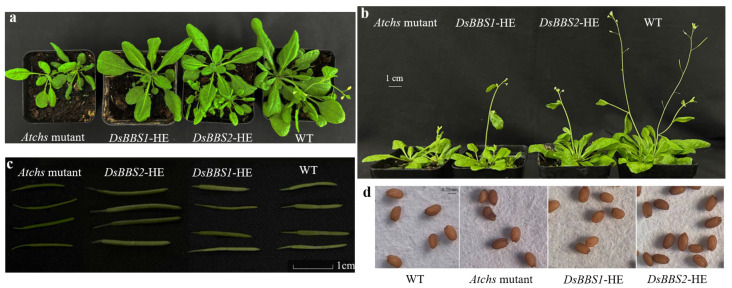
Comparative analysis of phenotypes among different *Arabidopsis* plants. (**a**,**b**) show healthy plants after 25 and 30 days of sowing, respectively. (**c**) Comparative analysis of pod. Plants were grown for about 60 days after sowing. Largest pod was taken from strongest inflorescence. (**d**) Comparative analysis of mature seeds. *DsBBS1*-HE and *DsBBS2*-HE, respectively, represent heterologous expressions of *DsBBS1* and *DsBBS2* in *Atchs* mutant.

**Figure 6 molecules-29-03682-f006:**
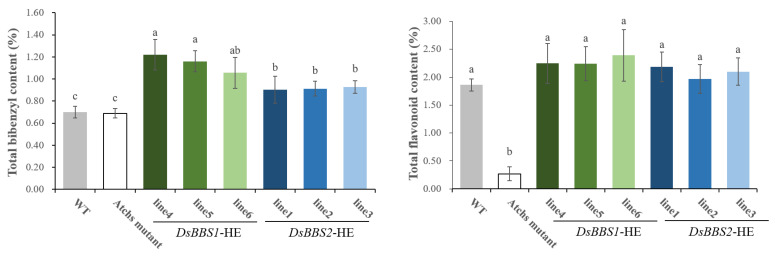
Total bibenzyl and flavonoid contents in *Arabidopsis* leaves. Sample was freeze-dried. Different letters indicate significant difference (*p* < 0.05; ANOVA). *DsBBS1*-HE and *DsBBS2*-HE, respectively, represent heterologous expressions of *DsBBS1* and *DsBBS2* in *Atchs* mutant.

## Data Availability

All data generated or analyzed during this study are included in this published article.

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
