# Peer review of "Unveiling the Catalytic Roles of DsBBS1 and DsBBS2 in the Bibenzyl Biosynthesis of Dendrobium sinense"

_molecules, 2024, doi:10.3390/molecules29153682_

Round 1

Reviewer 1 Report

Comments and Suggestions for Authors

In this manuscript under title „ Unveiling the catalytic roles of DsBBS1 and DsBBS2 in bibenzyl biosynthesis of Dendrobium sinense“ DsBBS1 and DsBBS2 function in D. sinense is extensively examined in this paper. Molecular docking simulation showed  minimal domain orientation discrepancies. DsBBS2 synthesised resveratrol with higher Vmax and lower Km than DsBBS1, showing increased catalytic efficiency.  The total flavonoid content in DsBBS1 and DsBBS2 was restored to wild-type levels, while the total bibenzyl content increased.

After this minor review, the manuscript should be accepted for publication.

Row 12 „Molecular simulation docking“ it is not written correctly.

Row 300 „respectively“ should be deleted.

Row 361 the sentence is not well written. It should be reformulated so that everything is clearly and correctly stated.

Rows 363 and 364 "total content of bibenzyl and flavonoid contents" is not well written, and "respectively" is unnecessary in my opinion, I don't know what it refers to.

Row 366 „ANOV analysis“ I don't know that something like this exists. It's probably a mistake.

Generally speaking, all essays are described in detail except for 4.7. Determination of total bibenzyl and flavonoid contents. So I think the authors should describe that as well.

Author Response

Thank you for the comments concerning our manuscript. The main corrections in the paper and the responds to the reviewer’s comments are as following:

Comments 1: Row 12 „Molecular simulation docking“ it is not written correctly.

Response 1: Thank you for pointing this out. We have revised “Molecular simulation docking” to “Molecular docking simulation”, line 13 and 86.

Comments 2: Row 300 „respectively“ should be deleted.

Response 2: Thank you for pointing this out. We have deleted “respectively”, line 296. Additionally, we added a more detailed description of experimental procedures.

Comments 3: Row 361 the sentence is not well written. It should be reformulated so that everything is clearly and correctly stated.

Response 3: Thank you for your feedback. This sentence has been modified, line 361-362, “Three-month-old Arabidopsis seedlings were used as the material to collect leaves for the analysis of the total bibenzyl and flavonoid contents”

Comments 4: Rows 363 and 364 "total content of bibenzyl and flavonoid contents" is not well written, and "respectively" is unnecessary in my opinion, I don't know what it refers to.

Response 4: Thank you for pointing this out. This sentence has been modified into “The gigantol (Yuanye, Shanghai, China) were used for generating standard curves to analysis the total bibenzyl content, while rutin (Yuanye, Shanghai, China) for the total flavonoid content.” (line 370-372)

Comments 5: Row 366 „ANOV analysis“ I don't know that something like this exists. It's probably a mistake.

Response 5: Thank you for pointing this out. This sentence has been modified, “The ANOVA analysis was conducted by GraphPad Prism software (P<0.05).”

Comments 6: Generally speaking, all essays are described in detail except for 4.7. Determination of total bibenzyl and flavonoid contents. So I think the authors should describe that as well.

Response 6: Thank you for your insightful input. A more detailed description of “Determination of total bibenzyl and flavonoid contents” has been added (line 363-375).

Reviewer 2 Report

Comments and Suggestions for Authors

In "Unveiling the catalytic roles of DsBBS1 and DsBBS2 in bibenzyl biosynthesis of Dendrobium sinense" by Liu et al. the authors identfied two type 3 PKS enzymes BBS1 & BBS2 from a Chinese traditional medicine plant, which catalyse the rate-limiting biosynthetic steps of bibebzyl compound biosynthesis, heterologously express them and test the activity of the expressed and purified enzymes using an end-point HPLC-based assay. The authors have done a notable amount of work, but the mauscript and the statements within belie a need for polishing and the authors have a notable tendency to make assertions that are a bit beyond the amount of data that was collected and as such notable revision is needed.

Minor points:
1) The English, while generally readable needs polishing and notable proof-reading.  For example, the first word in the sentence stating on line 35 is spelled incorrectly. As for polish, the line "Bibenzyl compounds, a class of polyketide ingredients" on line 48, "ingredients" is the incorrect term, "molecules" being more appropriate.

2) Figure 1 is not well organized.  I think I can understand what is the point of the figure, but I am not entirely sure.  It likely is clumsy due to the fact that it should be two figures.  One figure being a methodological overview which should be reduced to a simple ball and stick/ cartoon diagram and the other being illustrations of the binding sites and docking poses of the molecules in BBS1 & 2.  Probably the latter figure(s) should be moved to the supplemental.

3) line 93 - give the RMSD between the two structures

4) line 99 - just because a residue appears in a docking simulation, does not mean it is functionally relevant.  Attributions of function without experimental evidence must be removed.

5) Figure 5 A should not be in the main figure, move to supplemental. Explicitly identify what "mutant" is in the figure and caption.

6) Figure 6 needs a unit for its Y axis.  As it stands its unclear if its in mg or fold or something else.

7) The author's state "Indeed, the three-dimensional structural models of 230 DsBBS1 and DsBBS2 were almost identical (Figure 1e)." (line 229) But, this is obvious from the 95% similarity.  It would be quite shocking to see proteins with 95% sequence similarity and different structures.  The structures are almost identical because the proteins are almost identical.

8) Vmax and Km should be subscripted properly.

9) line 252 - two decimal places of accuracy for Km and Vmax are probably unwarranted for the number of measurements that seem to have been performed.  It is likely that the authors only achieved a single significant figure on these measurements.

10) line 256 - "The Km is defined…" this line is unneccesary as this is basic knowledge.

11) line 261 - "Based on molecular docking results, we speculate that the binding substrate active sites of DsBBS2 can enable the enzyme to change its conformation to achieve more optimal binding, thereby contributing catalysis" All enzymes change during the catalytic cycle to some degree (this is a consequence of binding the transition state more tightly than the substrate but still spending most of its time in a conformation attuned to substrate binding). The question is "how much change"?  The authors need to define exactly what changes (which residues and/or sections of the protein) and by how much (RMSD compared to the unbound/crystallographic state).  Numerical, quantified statements of the amount of change are needed here. Barring this, this set of conclusions needs to be altered.

12) line 300 - more details on the docking runs and set up are needed.

13) line 372 - assessment of catalytic efficiency requires unequivocal identification of the physiological substrate of the enzyme.  The authors assessed two and cannot rule out the possibility that they missed the correct substrate nor that both of these enzymes use the same one.

Comments on the Quality of English Language

The manuscript needs to be proof-read and some of the word choices, often synonyms, were rather questionable.

Author Response

Comments 1) The English, while generally readable needs polishing and notable proof-reading.  For example, the first word in the sentence stating on line 35 is spelled incorrectly. As for polish, the line "Bibenzyl compounds, a class of polyketide ingredients" on line 48, "ingredients" is the incorrect term, "molecules" being more appropriate.
Response 1: Thank you for your valuable feedback. The first word “Historically” in the sentence stating on line 34 has been revised. Also, “molecules” replaced “ingredients”. We have taken your suggestions into account and have made the necessary revisions to polish and proofread.

Comments 2) Figure 1 is not well organized.  I think I can understand what is the point of the figure, but I am not entirely sure.  It likely is clumsy due to the fact that it should be two figures.  One figure being a methodological overview which should be reduced to a simple ball and stick/ cartoon diagram and the other being illustrations of the binding sites and docking poses of the molecules in BBS1 & 2.  Probably the latter figure(s) should be moved to the supplemental.
Response 2: Thank you for your insightful comments regarding Figure 1. We have restructured the figure to enhance clarity and organization. Additionally, we have moved other result figures to the supplemental material of Figure S1.

Comments 3) line 93 - give the RMSD between the two structures
Response 3: Thank you for your specific inquiry about the RMSD between the two structures. We have conducted the calculation as per your suggestion and found that the RMSD is 0.14 (line 91). This value indicates a high degree of similarity between the structures, which supports the consistency and reliability of our findings.

Comments 4) line 99 - just because a residue appears in a docking simulation, does not mean it is functionally relevant.  Attributions of function without experimental evidence must be removed.
Response 4: In response to your comment, we have revised the manuscript to ensure that we only report the findings from the docking simulations. The text now states, 'ASP-136 was found in all docking models,' (line 100) which is a factual statement of the simulation results without any assumptions about its functional relevance.

Comments 5) Figure 5 A should not be in the main figure, move to supplemental. Explicitly identify what "mutant" is in the figure and caption.
Response 5: Thank you for your detailed feedback on Figure 5 A. We have taken your advice and relocated Figure 5 A to the supplemental material. Additionally, we have explicitly identified the 'mutant' in both the figure (Figure 5 and 6) and the caption to ensure clarity for the readers.

Comments 6) Figure 6 needs a unit for its Y axis.  As it stands its unclear if its in mg or fold or something else.
Response 6: Thank you for your attention to detail regarding Figure 6. we have added the unit for the Y-axis in Figure 6. The Y-axis now clearly indicates that the values are in 'percentage'.

Comments 7) The author's state "Indeed, the three-dimensional structural models of 230 DsBBS1 and DsBBS2 were almost identical (Figure 1e)." (line 229) But, this is obvious from the 95% similarity.  It would be quite shocking to see proteins with 95% sequence similarity and different structures.  The structures are almost identical because the proteins are almost identical.

Response 7: In light of your feedback, we have revised the manuscript to focus on the functional aspects of the proteins. “Indeed, the three-dimensional structural models of DsBBS1 and DsBBS2 were almost identical (Figure 1). The similarity of protein structure between DsBBS1 and DsBBS2 implies that they likely have similar functions, catalyzing the formation of bibenzyl compounds”. “DsBBS1 and DsBBS2 have the same catalytic activity, but their catalytic efficiency may vary.” line 227-236

Comments 8) Vmax and Km should be subscripted properly.
Response 8: In response to your feedback, we have corrected the subscripting of Vmax and Km throughout the manuscript. Line 19, 152, 155, 160, and 248.

Comments 9) line 252 - two decimal places of accuracy for Km and Vmax are probably unwarranted for the number of measurements that seem to have been performed.  It is likely that the authors only achieved a single significant figure on these measurements.
Response 9: In response to your feedback, we have reviewed the data and have decided to remove the specific numerical values for Km and Vmax from the Discussion part (line 248-250). The revised text now focuses on the result comparing.

Comments 10) line 256 - "The Km is defined…" this line is unneccesary as this is basic knowledge.

Response 10: In response to your feedback, we have removed the line 'The Km is defined…' from the manuscript.(line 250)

Comments 11) line 261 - "Based on molecular docking results, we speculate that the binding substrate active sites of DsBBS2 can enable the enzyme to change its conformation to achieve more optimal binding, thereby contributing catalysis" All enzymes change during the catalytic cycle to some degree (this is a consequence of binding the transition state more tightly than the substrate but still spending most of its time in a conformation attuned to substrate binding). The question is "how much change"?  The authors need to define exactly what changes (which residues and/or sections of the protein) and by how much (RMSD compared to the unbound/crystallographic state).  Numerical, quantified statements of the amount of change are needed here. Barring this, this set of conclusions needs to be altered.
Response 11: Thank you for your thorough and constructive feedback on the molecular docking results and the associated conclusions. we have modified the conclusions to reflect the fact that while the enzymatic activity is the same, the catalytic efficiency may differ due to variations in hydrogen bonding (line 253-256). Upon careful examination of the pre- and post-docking conformations, we observed no significant changes (RMSD=0) in the overall structure of DsBBS1 and DsBBS1.

Comments 12) line 300 - more details on the docking runs and set up are needed.
Response 12: Thank you for pointing this out. We have now provided detailed information on the parameters used, the software settings, and the steps followed during the docking process. (line 286-299)

Comments 13) line 372 - assessment of catalytic efficiency requires unequivocal identification of the physiological substrate of the enzyme.  The authors assessed two and cannot rule out the possibility that they missed the correct substrate nor that both of these enzymes use the same one.

Response 13: Thank you for your insightful comment. we have revised the conclusion to reflect a more cautious approach. In line 379-381, “The optimization of DsBBS2-HisTag expression conditions and the subsequent enzyme activity analysis revealed that DsBBS2 were able to catalyze the formation of bibenzyl compounds.” Furthermore, in the discussion section, we have added a new paragraph that addresses the limitations of our study (line 253-255).